# Genetic Characteristics and Phenotype of Korean Patients with Stickler Syndrome: A Korean Multicenter Analysis Report No. 1

**DOI:** 10.3390/genes12101578

**Published:** 2021-10-05

**Authors:** Soon-Il Choi, Se-Joon Woo, Baek-Lok Oh, Jinu Han, Hyun-Taek Lim, Byung-Joo Lee, Kwangsic Joo, Jun-Young Park, Ja-Hyun Jang, Min-Kyung So, Sang-Jin Kim

**Affiliations:** 1Department of Ophthalmology, Samsung Medical Center, Sungkyunkwan University School of Medicine, Seoul 06351, Korea; csi520dr@gmail.com; 2Department of Ophthalmology, Uijeongbu St. Mary’s Hospital, College of Medicine, The Catholic University of Korea, Uijeongbu 11765, Korea; 3Department of Ophthalmology, Seoul National University Bundang Hospital, Seoul National University College of Medicine, Seongnam 13620, Korea; sejoon1@snu.ac.kr (S.-J.W.); joo_man@hanmail.net (K.J.); yune1221@gmail.com (J.-Y.P.); 4Department of Ophthalmology, Seoul National University Hospital, Seoul National University College of Medicine, Seoul 03080, Korea; rocks529@gmail.com; 5Institute of Vision Research, Department of Ophthalmology, Gangnam Severance Hospital, Yonsei University College of Medicine, Seoul 06273, Korea; jinuhan@yuhs.ac; 6Department of Ophthalmology, Asan Medical Center, University of Ulsan College of Medicine, Seoul 05505, Korea; htlim@amc.seoul.kr (H.-T.L.); ozma805@gmail.com (B.-J.L.); 7Department of Laboratory Medicine and Genetics, Samsung Medical Center, Sungkyunkwan University School of Medicine, Seoul 06351, Korea; jahyun.jang@samsung.com (J.-H.J.); minkyung.so@samsung.com (M.-K.S.)

**Keywords:** Stickler syndrome, *COL2A1*, *COL11A1*, collagen, retinal detachment, genotype–phenotype correlation, myopia

## Abstract

Stickler syndrome is an inherited connective tissue disorder of collagen. There are relatively few reports of East Asian patients, and no large-scale studies have been conducted in Korean patients yet. In this study, we retrospectively analyzed the genetic characteristics and clinical features of Korean Stickler syndrome patients. Among 37 genetically confirmed Stickler syndrome patients, 21 types of gene variants were identified, of which 12 were novel variants. A total of 30 people had variants in the *COL2A1* gene and 7 had variants in the *COL11A1* gene. Among the types of pathogenic variants, missense variants were found in 11, nonsense variants in 8, and splice site variants in 7. Splicing variants were frequently associated with retinal detachment (71%) followed by missense variants. This is the first large-scale study of Koreans with Stickler syndrome, which will expand the spectrum of genetic variations of Stickler syndrome.

## 1. Introduction

Stickler syndrome was first described by Gunnar Stickler in 1965, and is clinically and genetically heterogenous as a hereditary connective tissue disorder of collagen [1]. The estimated incidence is 1:7500–1:10,000 depending on the study [2,3]. It is characterized by ophthalmic findings such as abnormal vitreous, myopia, and retinal detachment (RD) of variable degrees as well as skeletal, auditory, and orofacial abnormalities [4]. RD is a serious complication that can cause blindness in patients with Stickler syndrome, and is known to affect approximately 45–70% [5,6,7]. Moreover, Stickler syndrome is one of the major causes of primary RD in children [8].

Currently, six subtypes are identified according to their genotype, with 80–90% reported as type 1 Stickler syndrome (STL1, OMIM 108300) [9,10]. The remaining 10–20% are patients with type 2 Stickler syndrome (STL2, OMIM 604841), and it has been reported that patients with other subtypes are very rare. The known causative genes of Stickler syndrome are *COL2A1*, *COL11A1*, *COL11A2*, *COL9A1*, *COL9A2*, and *COL9A3*. Stickler syndrome caused by *COL2A1*, *COL11A1*, and *COL11A2* genes has a dominant inheritance, whereas *COL9A1*, *COL9A2*, and *COL9A3* genes show a recessive inheritance [4]. In the case of STL1 and STL2, hearing difficulties, orofacial abnormalities, and skeletal abnormalities are accompanied by ocular abnormalities. However, in the case of Stickler syndrome caused by *COL11A2*, it has been reported that the ocular phenotype and vitreous are normal [11]. Since Stickler syndrome caused by *COL9A1*, *COL9A2*, and *COL9A3* variants is very rare, most of the patients with Stickler syndrome are STL1 or STL2 patients if they have ocular abnormalities. One of the other subtypes, the ocular only variant of STL1 (OSTL1; MIM 609508) is known to be at high risk for RD with few accompanying systemic abnormalities [11,12]. It has been reported that OSTL1 is mainly caused by alternative splicing variants in exon 2 of the *COL2A1* gene [13,14], but also by variants in other areas than exon 2 [15].

Recently introduced NGS technology is very powerful and cost-effective in variant analysis, and is actively used in causative variant analysis of many diseases [16]. A number of results have been reported in Stickler syndrome patients [17,18,19]. Since the identification of Stickler syndrome, diagnosis has been made primarily by clinical aspects. However, genetic analysis has aided with the diagnosis of mild phenotype diseases and OSTL1, and has the potential to expand the phenotype spectrum of the disease.

So far, a large number of Stickler syndrome patients have been reported around the world [20], but relatively few studies have been done in the East Asian population [10]. In particular, several studies on Chinese and Japanese patients have been reported, but only a small case series of STL1 patients has been reported in Korea [14,15]. This multicenter study aimed to expand knowledge of the spectrum of genetic variations and clinical features in 37 Stickler syndrome patients with ocular involvement in Korea.

## 2. Materials and Methods

This was a multicenter retrospective study on patients who visited the ophthalmology clinic of five tertiary referral hospitals in Korea. The study conformed to the Declaration of Helsinki and was approved by the Institutional Review Board at Samsung Medical Center (2020-03-052-002), Seoul, Korea. The study was conducted on 37 individuals from 23 unrelated families.

All participants were diagnosed according to the criteria proposed by Snead et al. [21] and Rose et al. [22] and confirmed by genetic testing. According to the recommendation of Snead et al., if it is difficult to perform slit-lamp examination due to young age, a family history of RD or in sporadic cases of RD accompanied by systemic abnormalities, which were also taken into account.

Clinical data on best corrected visual acuity (BCVA), refractive error, slit-lamp biomicroscopic examination, fundus examination, and axial length measurement was obtained. Results of examinations on hearing difficulties, orofacial abnormalities, and skeletal abnormalities were also collected.

The genomic DNA was collected from peripheral blood of all available members of the family. DNA sequencing was analyzed through direct sequencing or by the method of gene panel analyzing and exome sequencing using NGS technology. Whole exome sequencing was performed according to each institution’s established methods: exome capture was performed using the SureSelect XT Human All Exon v5 or v6 kit (Agilent Technologies, Santa Clara, CA, USA) or the xGen Exome Research Panel v1.0 (Integrated DNA Technologies, Inc., Coralville, IA, USA) and sequencing was performed on a HiSeq 2500 (Illumina, San Diego, CA, USA) or NovaSeq 6000 (Illumina, San Diego, CA, USA). The hybridization capture-based gene panels used in each institution included known genes associated with vitreoretinopathy, which include *COL2A1*, *COL11A1*, *COL11A2*, *COL9A1*, *COL9A2*, and *COL9A3*. Sequencing was carried out using the Illumina NextSeq 550Dx platform (Illumina, San Diego, CA, USA) for targeted panel sequencing. The obtained DNA sequences of *COL2A1* were compared to the reference sequence NM_001844.4. For the *COL11A1* gene, NM_001854.3 was used as a reference sequence. The pathogenicity of the detected variants was classified according to the American College of Medical Genetics and Genomics (ACMG) guidelines [23].

## 3. Results

Of the total 37 patients (17 males) included in the analysis, 30 were Stickler syndrome type 1 (STL1) patients, while the remaining seven were Stickler syndrome type 2 (STL2) patients. There were no patients with Stickler syndrome caused by variants in other genes. The mean age of all patients was 19.6 years, with a range from 3 months to 56 years. The mean follow-up duration was 5.2 years. The related systemic disease was accompanied by hearing difficulties in 32.4% of patients, orofacial abnormalities in 45.9%, and skeletal abnormalities in 35.1% (Table 1).

Genetic tests on *COL2A1* and *COL11A1* genes were conducted at five tertiary centers: 18 patients were tested via direct sequencing, 10 patients with gene panel analysis, and 9 patients with whole exome sequencing. In 30 patients with *COL2A1* variants, 17 types of variants were identified, and in 7 patients with *COL11A1* variants, 4 types were identified. The overview of genetic characteristics of patients is presented in Table 2. According to the guidelines of the ACMG, there were 12 (32%) pathogenic variants and 15 (41%) likely pathogenic variants. Twelve novel variants were identified, nine of which were *COL2A1* variants, and the other three were *COL11A1* variants (Table 3).

There was one case (c.3106C>T) of the same variant in two unrelated families. The variants found in more than three patients were c.1693C>T, c.2003del, and c.3106C>T, all of which were variants in the *COL2A1* gene. The variants with two or more RD cases included c.1693C>T, c.2003del, c.3106C>T, c.3327+1G>C in *COL2A1*, and c.1630-2delA in *COL11A1* (Table 4). Of the total 37 patients (74 eyes), 18 (49%) showed RD (24 RD eyes, 32%). 

Among the variant types, missense variant was the most common with 11 patients, followed by 8 nonsense variants, 7 splicing, and 6 frameshift variants. In patients with splicing variants, 71% of patients, 50% of eyes had RD, and 29% of patients had bilateral RD (Table 5). In STL1 patients, 15 (50%) patients had RD, and 6 (20%) patients had bilateral RD. There were 3 (43%) RD patients in the STL2 group, but no bilateral RD patients.

Clinical features of eyes verified in the baseline study visit are shown in Table 6. The mean refractive error for all eyes was −8.23 ± 5.55 diopter (n = 66). Of the 53 phakic eyes with refractive error measurement, 39 eyes (73.58%) had myopia higher than −6 diopter. A total of 32 patients had axial length measurements taken, which showed a mean axial length of 27.59 ± 2.14 mm (range, 23.74 to 31.64 mm). The mean axial length was higher in patients with STL2 than those with STL1. Cataracts were found at the time of the baseline visit in 6 eyes (10%) in STL1 patients and 6 eyes (42.86%) in STL2 patients (Figure 1a). At the time of the baseline visit, 11 eyes (14.86%) had pseudophakia and 2 eyes (2.70%) aphakia. Of the remaining 64 eyes, except for the 10 eyes that had vitrectomy, 40 eyes (54.06%) were identified with vitreous anomaly (Figure 1b). Characteristic radial paravascular retinal degeneration was shown in 26 cases (25%). There were only 15 eyes (20.27%) without both degeneration of the typical radial paravascular pattern and non-typical lattice degenerations (Figure 1c). There were 21 cases (28.38%) of foveal hypoplasia at the time of baseline OCT and 17 cases (22.97%) of PVD progression from the beginning (Figure 1d). Figure 2 shows early PVD in case No. 5. She was diagnosed with partial PVD at the age of seven on her first visit to the hospital. PVD progressed rapidly until age 14, and RD occurred nine months after the last OCT exam. There were 10 eyes that had already been operated on for RD before the first visit. Of the remaining 64 eyes, 7 eyes had RD on their first visit to the hospital, and the other 7 eyes had developed RD during the follow-up period (Figure 1e).

## 4. Discussion

In this study, we report genetic analysis of 37 Korean patients with Stickler syndrome, which revealed 21 variants including 11 novel variants in *COL2A1* and *COL11A1*. We also report the first genetically confirmed STL2 cases in Korea. The proportion of patients with Stickler syndrome by subtype was similar to that in the previous report that 80–90% were STL1 and the remaining 10–20% were STL2 [4].

The diagnostic criteria mainly used in this study were those suggested by Rose et al. in 2005 [22]. If a patient was too young to perform an accurate slit-lamp examination, only those with a history of retinal detachment, or accompanying systemic features according to the criteria suggested by Snead et al. [21], were enrolled in this study. Boysen et al. have suggested that currently there is no general agreement on the diagnostic criteria for Stickler syndrome [7]. The two major diagnostic criteria were suggested before the era of active gene analysis due to the recent development of NGS technology, and do not include evaluation of the pathogenicity of gene variants according to the ACMG guidelines. In addition, as time passes after the main diagnostic criteria are presented, there are some differences between previously reported clinical features and recent results. In a report on Stickler syndrome over eight generations published in 2002, it was reported that radial perivascular retinal degeneration was present in 100% of patients with Stickler syndrome [12]. However, in a study using ultra-wide field fundus autofluorescence imaging in 2020, radial perivascular retinal degeneration was not found in more than 50% of cases [35]. In particular, retinal findings in Stickler syndrome showed an increase with age in a degenerative pattern, suggesting that young Stickler syndrome patients may not meet the diagnostic criteria. In a previous study, the presence of beaded vitreous was an essential factor in diagnosis of STL2, but none of the STL2 patients enrolled in this study had beaded vitreous. In previous studies conducted on Westerners, orofacial abnormality was one of the important diagnostic criteria but when analyzing Asians, it is argued that orofacial abnormality may be underestimated due to the characteristics of Asians’ appearance [10]. It is known that Stickler syndrome patients develop RD in the first two decades of life and have a high risk of blindness due to this [7]. Therefore, it is important to make an accurate diagnosis at an early age. However, the diagnostic criteria presented so far may have limitations in properly diagnosing Stickler syndrome in young patients or Asians for the reasons presented above.

In this study, we enrolled patients who met the main diagnostic criteria previously presented, and genetic diagnosis was performed in all patients. Therefore, the clinical phenotype of the patients shown in this study will help to re-establish the diagnostic criteria for patients with Stickler syndrome in the future. In this study, 27 patients were identified as PV or LPV, and 10 patients as VUS. All 10 patients with VUS pathogenicity were novel variant cases. In the case of a novel variant, there is no evidence score that can be received according to the previously reported reference [36]. Ten patients with VUS identified in this study were found to meet the previous clinical diagnostic criteria. Furthermore, since the diagnostic criteria were presented before the ACMG guidelines were published, judgment on pathogenicity of molecular evidence is different from now. Further studies will be needed in the future to determine whether a genetic variant interpreted as VUS pathogenicity is a finding that satisfies the diagnostic criteria.

The variants that were found in a large number of patients in this study were c.1693C>T, c.2003del, and c.3106C>T. According to a study of 107 East Asian STL1 patients analyzed by Wang et al., c.3106C>T, c.1833 + 1G>A, c.2710C>T, and c.1693C>T variants were reported in large numbers. It also reported that most of the variants were clustered in exon 42, exon 44, and exon 2 of the *COL2A1* gene [10]. To date, no disease-causing variant has been found in exon 20, 28, 53, and 54 in STL1 patients, but our study reported a variant of exon 28 for the first time.

In this study, splicing was the type of variant that caused a lot of cases of RD. These results are consistent with those of a previous study showing that splicing variants in East Asians had a more severe phenotype [10]. The incidence of RD in STL1 and STL2 patients differs from study to study [6,7,37]. According to a systematic review of 37 studies, there was no statistical difference in the risk of RD in STL1 (49%, 295 of 606 patients) and STL2 (38%, 13 of 34 patients) patients [7]. In this study, the number of STL2 patients was small, making statistical analysis difficult. The number of patients and eyes with RD were more frequent in STL1 than STL2 cases, and all patients with binocular RD were STL1.

In this study, at the first visit, only 10% of eyes had cataracts in the STL1 group, but STL2 patients had a higher rate (43%). In a meta-analysis of East Asian STL1 patients, the cataract comorbidity rate was reported to be 22.1% [10], and according to the STL2 study conducted in the UK [6], cataracts were reported in 64%. Between the two types of Stickler syndrome patients, the comorbidity of hearing difficulties was also higher in STL2 patients, and skeletal abnormalities were higher in STL1. This pattern was similar to the previous report that there was a difference in the accompanying hearing difficulties of STL1 and STL2 [4] patients. However, a statistical difference could not be confirmed due to the small number of patients. Additional studies with a larger number of patients will be needed to analyze the differences in clinical features between the two groups, STL1 and STL2.

PVD is known to commonly occur in patients with Stickler syndrome and is particularly considered as an early suggestive sign of STL [38]. In this study, 22.97% of all patients had PVD at baseline. Considering that at baseline visit the average patient was aged in their 20s, early onset PVD was found to occur frequently. In Figure 2, it was confirmed that the patient had PVD from the age of 7 and the condition progressed rapidly until the age of 14. As seen in Figure 2d, it could be expected that complete PVD would occur soon after the time of examination, and RD occurred 9 months later. According to a systematic review of ocular complications, RD in STL1 patients occurs frequently in the second decade [7]. Early PVD appears to be correlated with RD onset at an early age. Therefore, taking OCT scans from children and evaluating PVD progression seems to be very helpful in predicting the occurrence of RD.

The limitations of this study are that it was conducted as a retrospective analysis, and that a relatively small number of STL2 patients were enrolled. However, the genetic results were confirmed in all patients and analyzed according to the recently presented ACMG criteria, and accurate analysis was attempted by enrolling patients who met the existing diagnostic criteria. This is the first large-scale study targeting Korean patients with Stickler syndrome, and further research will be possible by expanding the gene spectrum.

## 5. Conclusions

This multi-center study was able to broaden the genetic variation spectrum of *COL2A1* and *COL11A1* genes in Korean patients with Stickler syndrome. For more accurate diagnosis in patients with ocular-only variants, patients with mild phenotypes, and infants and children, genetic testing will be useful for the proper diagnosis.

## Figures and Tables

**Figure 1 genes-12-01578-f001:**
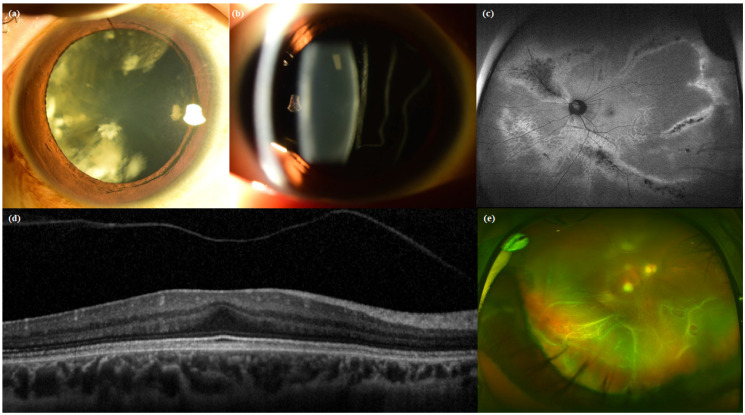
Ocular features of Stickler syndrome: (**a**) quadrantic lamellar cataract, (**b**) membranous vitreous, (**c**) radial paravascular pigmentation, (**d**) foveal hypoplasia and early PVD shown at the age of 10 in OCT, (**e**) total retinal detachment caused by multiple small retinal holes.

**Figure 2 genes-12-01578-f002:**
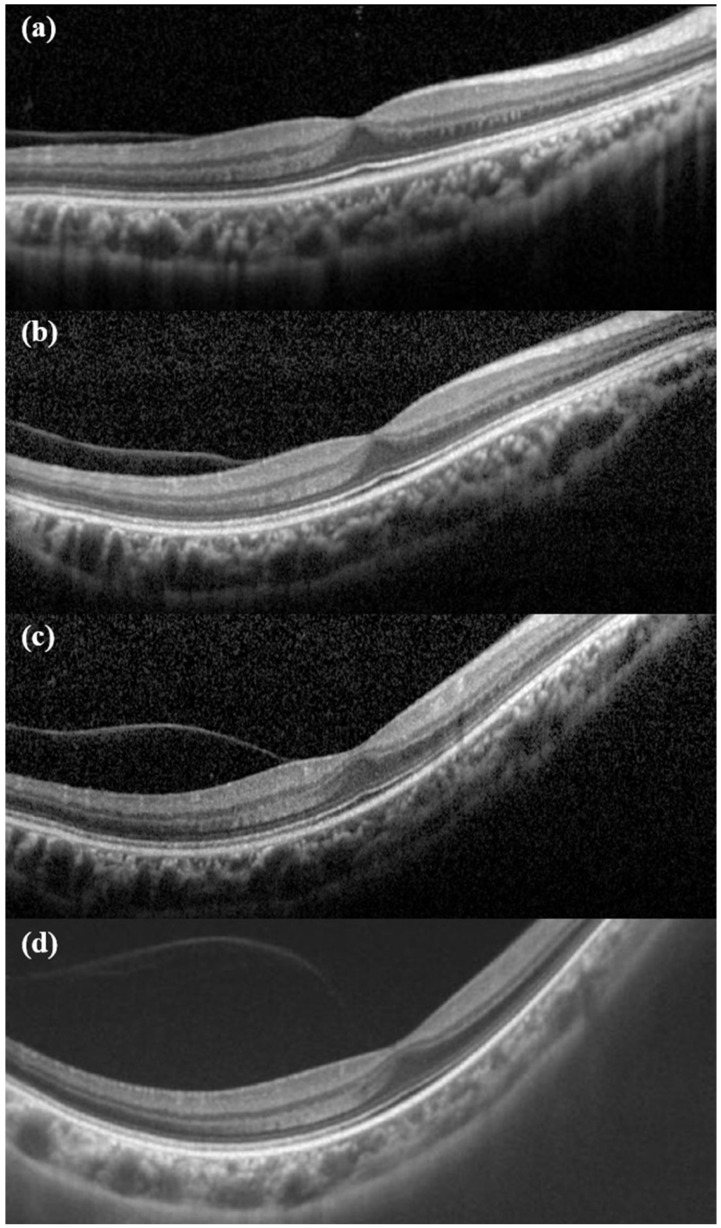
Chronological changes of early PVD. (**a**) First OCT exam at age 7. At that time, the refractive error was −10 diopter. (**b**) After 3 years, it can be seen that PVD has partially progressed. (**c**) At the age of 13, the refractive error was −11.75D. Compared with (**a**), it can be seen that the contour of eyeball is more myopic. (**d**) At the age of 14, PVD showed further progress. Retinal detachment occurred 9 months later.

**Table 1 genes-12-01578-t001:** Patient demographics and systemic findings.

	Total (n = 37)	Stickler Syndrome Type 1 (n = 30)	Stickler Syndrome Type 2 (n = 7)
Male:Female	17:20	15:15	2:5
Mean (range) age at first visit (yrs)	19.6 (0.25–56)	18.7 (0.25–56)	23.4 (0.25–56)
Mean (range) follow-up duration (yrs)	5.2 (0–19)	5.6 (0–19)	3.4 (0–12)
Associated systemic features (%)			
Hearing difficulties †	12 (32.4)	9 (30.0)	3 (42.9)
Orofacial abnormalities	17 (45.9)	14 (46.7)	3 (42.9)
Cleft palate	14 (37.8)	12 (40.0)	2 (28.6)
Others ‡	4 (10.8)	3 (10.0)	1 (14.3)
Skeletal abnormalities	13 (35.1)	13 (43.3)	0 (0)
Spondyloepiphyseal dysplasia	5 (13.5)	5 (16.7)	0 (0)
Others *	11 (29.7)	11 (36.7)	0 (0)

† High-frequency sensorineural hearing loss, hypermobile tympanic membranes; ‡ malar hypoplasia, broad or flat nasal bridge, micrognathia; * scoliosis, osteoarthritis before age 40, hyper-extensibility, talipes equinovarus, pectus carinatum, pectus.

**Table 2 genes-12-01578-t002:** Overview of genetic characteristics of Stickler syndrome patients.

ID	Family	Gene Symbol	Exon/Intron	Nucleotide Change	Protein Change	Variant Type	ACMG Classification	References
1	A	*COL2A1*	Ex42	c.2862C>T	p.(Gly954=)	Sn	LPV	Richards et al. [9]
2	B	*COL2A1*	Ex40	c.2678dup	p.(Ala895Serfs*49)	F	PV	Hoornaert et al. [24] Barat-Houari et al. [25] Yoon et al. [15]
3	C	*COL2A1*	IVS47	c.3327+1G>C	p.(?)	S	LPV	Yoon et al. [15]
4	C	*COL2A1*	IVS47	c.3327+1G>C	p.(?)	S	LPV	Yoon et al. [15]
5	D	*COL2A1*	Ex31	c.2003del	p.(Pro668Leufs*120)	F	PV	Novel
6	D	*COL2A1*	Ex31	c.2003del	p.(Pro668Leufs*120)	F	PV	Novel
7	D	*COL2A1*	Ex31	c.2003del	p.(Pro668Leufs*120)	F	PV	Novel
8	E	*COL2A1*	Ex51	c.3867C>A	p.(Cys1289*)	N	PV	Novel
9	E	*COL2A1*	Ex51	c.3867C>A	p.(Cys1289*)	N	PV	Novel
10	F	*COL2A1*	Ex52	c.4044G>C	p.(Trp1348Cys)	M	VUS	Novel
11	G	*COL11A1*	Ex49	c.3703G>A	p.(Ala1235Thr)	M	VUS	Novel
12	G	*COL11A1*	Ex49	c.3703G>A	p.(Ala1235Thr)	M	VUS	Novel
13	H	*COL11A1*	IVS15	c.1630-2delA	p.(?)	S	LPV	Martin et al. [26]
14	H	*COL11A1*	IVS15	c.1630-2delA	p.(?)	S	LPV	Martin et al. [26]
15	I	*COL2A1*	IVS40	c.2680-3C>G	p.(?)	I	VUS	Novel
16	J	*COL2A1*	Ex42	c.2862C>T	p.(Gly954=)	Sn	LPV	Richards et al. [9]
17	K	*COL2A1*	IVS45	c.3165+1G>A	p.(?)	S	VUS	Novel
18	L	*COL2A1*	IVS13	c.870+1G>A	p.(?)	S	LPV	Richards et al. [9]
19	M	*COL2A1*	Ex9	c.625C>T	p.(Arg209*)	N	PV	Ahmad et al. [27]
20	N	*COL2A1*	Ex51	c.3598G>C	p.(Gly1200Arg)	M	LPV	Novel
21	O	*COL2A1*	Ex48	c.3394del	p.(His1132Thrfs*95)	F	LPV	Novel
22	P	*COL2A1*	Ex28	c.1844del	p.(Gly615Alafs*14)	F	LPV	Novel
23	Q	*COL2A1*	Ex26	c.1693C>T	p.(Arg565Cys)	M	LPV	Richards et al. [28], Sun et al. [29], Wang et al. [30], Zhou et al. [31]
24	Q	*COL2A1*	Ex26	c.1693C>T	p.(Arg565Cys)	M	LPV	Richards et al. [28], Sun et al. [29], Wang et al. [30], Zhou et al. [31]
25	Q	*COL2A1*	Ex26	c.1693C>T	p.(Arg565Cys)	M	LPV	Richards et al. [28], Sun et al. [29], Wang et al. [30], Zhou et al. [31]
26	Q	*COL2A1*	Ex26	c.1693C>T	p.(Arg565Cys)	M	LPV	Richards et al. [28], Sun et al. [29], Wang et al. [30], Zhou et al. [31]
27	Q	*COL2A1*	Ex26	c.1693C>T	p.(Arg565Cys)	M	LPV	Richards et al. [28], Sun et al. [29], Wang et al. [30], Zhou et al. [31]
28	R	*COL2A1*	Ex44	c.3106C>T	p.(Arg1036*)	N	PV	Zhou et al. [31], Richards et al. [32], Savasta et al. [33]
29	R	*COL2A1*	Ex44	c.3106C>T	p.(Arg1036*)	N	PV	Zhou et al. [31], Richards et al. [32], Savasta et al. [33]
30	R	*COL2A1*	Ex44	c.3106C>T	p.(Arg1036*)	N	PV	Zhou et al. [31], Richards et al. [32], Savasta et al. [33]
31	R	*COL2A1*	Ex44	c.3106C>T	p.(Arg1036*)	N	PV	Zhou et al. [31], Richards et al. [32], Savasta et al. [33]
32	S	*COL11A1*	IVS51	c.3816+2dup	p.(?)	S	VUS	Novel
33	T	*COL2A1*	Ex23	c.1493G>A	p.(Gly498Asp)	M	VUS	Novel
34	U	*COL11A1*	Ex29	c.2308_2316del	p.(Val770_Gly772del)	IFD	VUS	Novel
35	U	*COL11A1*	Ex29	c.2308_2316del	p.(Val770_Gly772del)	IFD	VUS	Novel
36	V	*COL2A1*	Ex11	c.737G>A	p.(Gly246Asp)	M	VUS	Lee et al. [34]
37	W	*COL2A1*	Ex44	c.3106C>T	p.(Arg1036*)	N	PV	Zhou et al. [31], Richards et al. [32], Savasta et al. [33]

M = missense; Sn = synonymous; S = splicing (canonical splicing site); N = nonsense; F = frame shift; IFD = in-frame deletion; I = intron (non-canonical splice site).

**Table 3 genes-12-01578-t003:** Classification of identified variants by ACMG guidelines.

		Total (n = 37)	*COL2A1* (n = 30)	*COL11A1* (n = 7)
ACMG classification	PV (%)	12 (32)	12 (40)	0 (0)
	LPV (%)	15 (41)	13 (43)	2 (29)
	VUS (%)	10 (27)	5 (17)	5 (71)
Novel variants (%)		17 (46)	12 (40)	5 (71)

**Table 4 genes-12-01578-t004:** The distribution of variants according to exon/intron and pattern of retinal detachment.

Gene	Exon/Intron	Nucleotide Change	No. of Patients	No. of RD Patients	No. of Bilateral RD Patients	No. of RD Eyes
*COL2A1*	Ex9	c.625C>T	1	1		1
*COL2A1*	Ex11	c.737G>A	1	1		1
*COL2A1*	IVS13	c.870+1G>A	1	1		1
*COL2A1*	Ex23	c.1493G>A	1			
*COL2A1*	Ex26	c.1693C>T	5	4	3	7
*COL2A1*	Ex28	c.1844del	1			
*COL2A1*	Ex31	c.2003del	3	3	1	4
*COL2A1*	Ex40	c.2678dup	1			
*COL2A1*	IVS40	c.2680-3C>G	1			
*COL2A1*	Ex42	c.2862C>T	2			
*COL2A1*	Ex44	c.3106C>T	5	2		2
*COL2A1*	IVS45	c.3165+1G>A	1			
*COL2A1*	IVS47	c.3327+1G>C	2	2	2	
*COL2A1*	Ex48	c.3394del	1			
*COL2A1*	Ex51	c.3598G>C	1			
*COL2A1*	Ex51	c.3867C>A	2			
*COL2A1*	Ex52	c.4044G>C	1	1		1
*COL11A1*	IVS15	c.1630-2delA	2	2		2
*COL11A1*	Ex29	c.2308_2316del	2	1		1
*COL11A1*	Ex49	c.3703G>A	2			
*COL11A1*	IVS51	c.3816+2dup	1			

**Table 5 genes-12-01578-t005:** The distribution of variant types and related retinal detachment occurrences.

Variant Type	No. of RD Patients (%)	No. of Bilateral RD Patients (%)	No. of RD Eyes (%)
Missense (n = 11)	6 (55)	3 (27)	9 (41)
Synonymous (n = 2)	0 (0)	0 (0)	0 (0)
Splicing (n = 7)	5 (71)	2 (29)	7 (50)
Nonsense (n = 8)	3 (38)	0 (0)	3 (19)
Frame shift (n = 6)	3 (50)	1 (17)	4 (33)
In-frame deletion (n = 2)	1 (50)	0 (0)	1 (25)
Intron (n = 1)	0 (0)	0 (0)	0 (0)

**Table 6 genes-12-01578-t006:** Baseline clinical features of eyes.

Baseline Status		Total (N = 74)	Stickler Type 1 (N = 60)	Stickler Type 2 (N = 14)
Mean BCVA (logMAR)		0.40 ± 0.56 (n = 60)	0.39 ± 0.57 (n = 48)	0.45 ± 0.57 (n = 12)
Mean (range) Refractive error (D)		−8.23 ± 5.55 (−23 ± 0.125, n = 66)	−8.12 ± 5.49 (−23 ± 0, n = 52)	−8.62 ± 5.98 (−20 ± 0.125, n = 14)
Mean (range) Axial length (mm)		27.59 ± 2.14 (23.74–31.64, n = 32)	27.39 ± 2.17 (23.74–31.64, n = 26)	28.43 ± 1.93 (26.39–31.26, n = 6)
Mean (range) age at the time of axial length measurement (year)		24.75 ± 17.38 (4–56, n = 32)	23.23 ± 17.48 (4–56, n = 26)	31.33 ± 16.74 (10–45, n = 6)
Lens	Phakia, with cataract (%)	12 (16.2)	6 (10.0)	6 (42.9)
	Phakia, without cataract (%)	46 (62.2)	42 (70.0)	4 (28.6)
	Pseudophakia (%)	11 (14.9)	6 (10.0)	5 (35.7)
	Aphakia (%)	2 (2.7)	2 (3.3)	0 (0)
	Anophthalmos (%)	2 (2.7)	2 (3.3)	0 (0)
Vitreous	Membranous vitreous (%)	35 (47.3)	31 (51.7)	4 (28.6)
	Beaded vitreous (%)	5 (6.8)	5 (8.3)	0 (0)
	Initial PVD (%)	17 (23.0)	13 (21.7)	4 (28.6)
	N/A (%)	10 (13.5)	9 (15.0)	1 (7.1)
Retina	Paravascular pigmented atrophic lesion (%)	26 (35.1)	21 (35.0)	4 (28.6)
	Lattice degeneration (%)	54 (73.0)	45 (75.0)	9 (64.3)
	Foveal hypoplasia (%)	21 (28.4)	16 (26.7)	5 (35.7)
	N/A (%)	3 (4.1)	3 (5.0)	0 (0)

## Data Availability

Not applicable.

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
