# Peer review of "Genetic Characteristics and Phenotype of Korean Patients with Stickler Syndrome: A Korean Multicenter Analysis Report No. 1"

_genes, 2021, doi:10.3390/genes12101578_

Round 1

Reviewer 1 Report

This retrospective study reports the genotypes of a small number of patients with Stickler Syndrome.  Mutations were identified in two (Col11a1 and Col2a1) of the six causative genes, that were shown to cause this phenotype two and three decades ago respectively.  Although some of the clinical images are nice, the key issue is lack of novelty, reflecting the fact that a large number of Stickler-causing mutations have previously been reported over the past 20+ years.

It was notable that the brief review in the introduction omits mention of most of the causative genes, inheritance patterns or known genotype-phenotype correlations. Equally, the phenotypic data provided are very 'eye-centric'.

Author Response

Regarding the report of Reviewer 1 that this study mainly focuses on ophthalmic contents, we will give the following answer.

1) As pointed out by reviewer 1, the causative gene, inheritance pattern, and genotype-phenotype correlation of Stickler syndrome were additionally described in the introduction. 

(Please see the revised manuscript. In this revised manuscript, the modified part is marked in red.)

As mentioned in the revised manuscript, most eye problems in patients with Stickler syndrome are related to the COL2A1 and COL11A1 genes. 

In addition, the patients who participated in this study were patients who visited the ophthalmology clinic of 5 tertiary institutions.

2) So we added the phrase "with ocular involvement" at the end of the introduction of the revised manuscript.

3) In the Materials and Methods section, we revealed that the patients participating in this study were “patients who visited the ophthalmology clinic”.
4) In the first paragraph of the Result section, it was stated that no other causative gene was identified as a result of the genetic test.
5) In Table 1, data on detailed musculoskeletal and orofacial abnormalities are additionally displayed.

As above, the details of this study have been modified so that it can be a more comprehensive study on Stickler syndrome. 

Also, we wanted to make it clearer that this study was a study of patients with Stickler syndrome with ocular involvement.

We hope this can be an answer to your thoughtful remarks. 

Thank you.

Reviewer 2 Report

Choi et al. retrospectively analyzed the genetic characteristics and clinical features of Korean Stickler syndrome patients. A total of 30 people had mutations in the COL2A1 gene and 7 had mutations in the COL11A1 gene. This is the first large-scale study of Koreans with Stickler syndrome, which will expand the spectrum of genetic variations of Stickler syndrome.

Minor revision:

  1. Please note the following preferred terminology:
    • variant instead of mutation
    • pathogenic variant to denote a disease-causing variant
  2. Nomenclature for gene variants: c.2678dup instead of c.2678dupC; c.3816+2dup instead of c.3816+2dupT.
  3. Methods section must be improved. NGS analysis including whole exome sequencing, panel targeted, and Sanger sequencing must be described. 

Author Response

Please see the revised manuscript. In the revised manuscript, the modified part is marked in red.

Response 1) The term was modified to variant instead of mutation, and the pathogenic variant was used to denote a disease-causing variant.

Response 2) Changed to c.2678dup and c.3816+2dup instead of c.2678dupC and c.3816+2dupT.

Response 3) Details related to NGS analysis are additionally described in the Materials and Methods section.

We hope this can be an answer to your thoughtful remarks. 

Thank you.

Round 2

Reviewer 1 Report

-